# Antinociceptive Effect of Single Components Isolated from *Agrimonia pilosa* Ledeb. Extract

**Jing Hui Feng [1,2], Hee Jung Lee [1,2], Set Byeol Kim [3], Jeon Sub Jung [1,2], Soon Sung Lim [1,3]**  **and Hong Won Suh [1,2,*]**

[1] Institute of Natural Medicine, Hallym University, Chuncheon 200-702, Korea
[2] Department of Pharmacology, College of Medicine, Hallym University, Chuncheon 200-702, Korea
[3] Department of Food Sciences and Nutrition, College of Natural, Health, and Life Sciences, Hallym University, Chuncheon 200-702, Korea
[*] Correspondence: hwsuh34@gmail.com; Tel.: +82-33-248-2614; Fax: +82-33-248-2612

**Abstract:** *Agrimonia pilosa* Ledeb. produces an antinociceptive effect in ICR mice in both chemically induced and thermal pain models. In the present study, we examined the antinociceptive effects of single components isolated from *Agrimonia pilosa* Ledeb. (AP) extract in ICR mice. Three active compounds isolated from AP, including rutin, luteolin-7-*O*-glucuronide, and apigenin-7-*O*-glucuronide, were isolated and identified by comparing EI-MS, 1H-, 13C-NMR, and UV. We studied the antinociceptive effects of three single components administered orally at doses of 10 and 20 mg/kg in monosodium urate (MSU)-treated pain model as measured by von Frey test. Among these compounds, apigenin-7-*O*-glucuronide was more effective in the production of antinociceptive effects. We further characterized the antinociceptive effects and possible mechanisms of apigenin-7-*O*-glucuronide in writhing and formalin tests. Oral administration of Apigenin-7-*O*-glucuronide caused a reduction in the number of writhing and effectively reduced the pain behavior observed during the second phase of the formalin test in a dose-dependent manner. In addition, the pretreatment of yohimbine instead of naloxone or methysergide attenuated apigenin-7-*O*-glucuronide-induced antinociception in the writhing test. Moreover, apigenin-7-*O*-glucuronide caused reduction in the expression of p-P38, p-CREB, and p-mTOR induced by formalin injection. Our results indicate that apigenin-7-*O*-glucuronide shows an antinociceptive effect in various pain models. In addition, spinal $\alpha_2$-adrenergic receptors appear to be involved in the production of antinociception induced by apigenin-7-*O*-glucuronide. Furthermore, the antinociceptive effect of apigenin-7-*O*-glucuronide appears to be mediated by reduction in the expression of p-P38, p-CREB and p-mTOR levels in the spinal cord.

**Keywords:** apigenin-7-*O*-glucuronide; antinociception; $\alpha_2$-adrenergic receptors; spinal signal molecules

## 1. Introduction

*Agrimonia pilosa* Ledeb. (AP) is a flowering plant in the Rosacea family; it has been used traditionally for the treatment of abdominal pain, sore throat, headaches, bloody discharge, parasitic infections, and eczema in Korea and other Asian countries [1]. It has been reported that AP possesses antitumor [2,3], anti-viral [4,5], anti-oxidant [6], antimicrobial [7], anti-hyperglycemic activity [8], and anti-inflammation [9,10].

We have previously reported that AP extract produces a significant antinociceptive effect in chemically induced pain models, such as writhing and formalin tests as well as thermal pain models such as tail-flick and hot-plate tests, indicating the AP extract alleviate chemical and thermal nociception [11]. In addition, we found that the antinociceptive effect induced by AP extract appears to be mediated

by spinally located $\alpha$2-adrenergic receptors [11]. Although we have shown AP extract is effective in relieving the pain in writhing and formalin tests, it has not been well characterized which components are responsible for the production of antinociception. Therefore, the present study was designed to examine the possible effects of several single components isolated from AP extract in various pain models.

The p-38 and cAMP response element-binding proteins (CREB) are reported to be considered as important factors to study the effect of antinociceptive drugs in the regulation of pain. Meotti et al. [12] suggests that the mechanism for antinociceptive action of myricitrin in response to cytokines involves a blockage on the p-38 (MAPK) pathway. Additionally, Wang et al. [13] indicates that regulation of CREB signaling may be the targets for the antinociceptive effects of oxymatrine on a chronic neuropathic pain model. Consistently, pharmacological inhibition of the m-TOR turned out to have significant antinociceptive effects in several experimental models of inflammatory and neuropathic pain, such as formalin-induced inflammation model [14–16]. Thus, in the present study, we have tried to find the effect of a single component on the phosphorylation of m-TOR, p-38, and CREB proteins in a the formalin pain model.

## 2. Materials and Methods

### 2.1. Plant Materials

*Agrimonia pilosa* Ledeb. (AP) was purchased from a local market in Chuncheon, Republic of Korea. A voucher sample (RIC-2015-63) has been deposited at the Center for Efficacy Assessment and Development of Functional Foods and Drugs, Hallym University, Chuncheon. The specimens were authenticated by Emeritus Professor H. J. Chi, Seoul National University, Republic of Korea.

### 2.2. Extraction and Fractionation of Agrimonia pilosa Ledeb. Extract

The dried AP (10 kg) was extracted twice in 100 L of 50% ethanol (EtOH) for 5 h. The filtrate was combined and evaporated to dryness, leading to the 50% EtOH extract (yield: 13.5%). The extract was suspended in distilled water and successively partitioned with ethyl acetate (EtOAc), *n*-butanol (*n*-BuOH), and $H_2O$ to afford EtOAc fraction (110.02 g) and *n*-BuOH fraction (131.96 g), as well as water fraction (381.32 g).

### 2.3. Isolation of Single Components from Agrimonia pilosa Ledeb. Extract

The *n*-BuOH fraction was subjected to chromatography on a Diaion HP-20 resin column using water-MeOH (70:30 to 0:100) as the eluent, thereby yielding ten sub-fractions. The three compounds (compound 1~3) were isolated from sub-fraction 5 (0.3 g) on Sephadex LH-20 column (90 cm × 3 cm i.d.) through elution with aqueous MeOH gradient (40%, 50%, 60%, 70%, 80%, 90%, and 100% each). Fraction was collected 20 mL of volume and confirmed by HPLC.

### 2.4. Experimental Animals

Male ICR mice weighing 20–25 g were purchased from MJ Co., Seoul, Korea and divided into several groups, each group consisted of 5 mice. Animals were kept in a room maintained at 22 ± 0.5 °C with a 12:12 h light dark cycle and unlimited access to food and water. The animals were allowed to adapt to the laboratory for at least two hours before testing and were only used once. All experiments were carried out between 10:00 and 17:00. The study protocol was approved by the Hallym University Animal Care and Use Committee (Registration Number: Hallym 2009-05-01) in accordance with the 'Guide for Care and Use of Laboratory Animals' published by the National Institutes of Health and the ethical guidelines of the International Association for the Study of Pain.

### 2.5. Production of MSU-Induced Gout Pain Model

Monosodium urate (MSU) crystals were prepared as described previously [17]. 1 g of uric acid (Sigma, USA) was dissolved and heated in 180 mL of NaOH (0.01 M) at 70 °C and adjusted the pH between 7.1 and 7.2. The solution was filtered and incubated at room temperature, with slow and continuous stirring, for 24 h. MSU crystals were kept sterile, washed with ethanol, dried, autoclaved, and resuspended in phosphate-buffered saline (PBS) by sonication. The MSU crystal suspension (0.5 mg/10 μL) was injected intraarticularly (IA) into the right ankle joint. We used Microliter #705 syringes (Hamilton, USA) with 27-gauge needles for all IA injections. The mice were tested and orally administered of drugs at 24 h after the injection.

### 2.6. Von-Frey Test

Antinociception and mechanical allodynia were assessed through the up-down method applying the von Frey test [18]. Mice were placed in a clear glass cell individually atop a wire mesh grid to acclimatize the testing for 30 min. Then, von Frey filaments (North Coast Medical, Inc., Gilroy, CA, USA) were applied to the plantar surface of the right hind paw until the fiber bends. Brisk withdrawal of the hind paw during or immediately after application was considered a positive response. Threshold values were assessed in 8 groups of mice (normal, control, rutin 10 mg/kg, rutin 20 mg/kg, luteolin-7-*O*-glucuronide 10 mg/kg, luteolin-7-*O*-glucuronide 20 mg/kg, apigenin-7-*O*-glucuronide 10 mg/kg, and apigenin-7-*O*-glucuronide 20 mg/kg). After baseline values were obtained, the drugs were administrated and mechanical threshold evaluated at 30, 60, and 120 min later.

### 2.7. Acetic Acid-Induced Writhing and Intraplantar Formalin Tests

The writhing and formalin tests were used to assess the antinociceptive effect of the AP extract. As described by Koster et al. [19], the mice were injected i.p. with 1% acetic acid and then were placed in an acrylic observation chamber immediately. The number of writhing was counted over the following 30 min. A writhe was defined as an abdominal contraction of the forelimbs and elongation of the body. As described by Hunskaar and Hole [20], 10 μL of 5% formalin was injected subcutaneously (s.c.) into the left hind paw. Then the mice were placed immediately in an acrylic observation chamber. Quantification of nociception was based on the animal behaviors (licking, shaking, or biting the injected paw) during a period of 40 min using a manually stopwatch. The first phase (acute pain) was considered 0–5 min, following an interphase (6–20 min) the second phase (inflammatory) continued from 20th to 40th after injection. Animals were orally pretreated with apigenin-7-*O*-glucuronide (10 or 20 mg/kg) with 30 min prior to performing the acetic acid-induced writhing test or formalin test.

### 2.8. Spinal Pretreatment of Antagonists

At first, mice were pretreated i.t. with either saline, methysergide (0.01 μg/5 μL), yohimbine (0.01 μg/5 μL), or naloxone (0.01 μg/5 μL) 5 min before oral administration of vehicle as a control or a fixed dose of apigenin-7-*O*-glucuronide (20 mg/kg). And then the writhing response was measured 30 min after the treatment with either vehicle or apigenin-7-*O*-glucuronide.

### 2.9. Protein Extraction and Western Blot

Mice were decapitated; and then the lumbar section of the spinal cord was dissected immediately. Tissue samples were washed twice with cold Tris-buffered saline (20 mmol/L Trizma base and 137 mmol/L NaCl, pH 7.5), immediately frozen, and stored in the ultra-lower temperature refrigerator (−80 °C) until assay. The dissected spinal tissues were lysed with sodium dodecyl sulfate lysis buffer (62.5 mmol/L Trizma base, 2% *w/v* sodium dodecyl sulfate, 10% glycerol) containing 0.1 mmol/L $Na_3VO_4$, 3 mg/mL aprotinin, and 20 mmolL NaF. The sample was then centrifuged at 13,000 rpm for 15 min at 4 °C, and the supernatant was retained. Protein concentrations were evaluated with the Bradford method (Bio-Rad Laboratories, Hercules, CA, USA) using bovine serum albumin as the

standard. The samples were boiled after adding bromophenol blue (0.1% *w/v*). Equal amounts of protein were resolved by 6–10% SDS-polyacrylamide gel electrophoresis system and transferred to a polyvinylidene difluoride membrane (Millipore, Bedford, MA, USA). After blocking (2 h at room temperature) with 5% skim milk in Tris-buffered saline containing 20% Tween-20 (TBST; 10 mM Trizma base, pH 8.0, 150 mM NaCl, and 20% Tween 20), the membranes were immunoblotted with antibodies p-mTOR (Abcam, Cambridge, UK, 1:1000), p-P38 (Abcam, Cambridge, UK, 1:1000), p-CREB (Abcam, Cambridge, UK, 1:1000) and β-actin (Cell Signaling Technology, USA, 1:1000) in a blocking buffer for overnight at 4 °C. Then, the membranes were washed 4 times with TBST for 20 min and incubated with the anti-rabbit IgG-horseradish peroxidase conjugated secondary antibody (Enzo Life Sciences, Danvers, MA, USA, 1:4000) in blocking buffer at room temperature for 1 h. After washing the membranes with TBST for 20 min 4 times, the antibody-antigen complexes were detected using the ECL system and exposed to Luminescent Image Analyzer (LAS-4000, Fuji Film Co., Tokyo, Japan) for the detection of light emission. p-mTOR, p-P38, p-CREB, and β-actin band densities were evaluated from the respective band densitometry. The Multi-Gauge Version 3.1 (Fuji Film Co., Tokyo, Japan) was used to analyze the intensity of expression. These values were expressed as the percentage of the control tested protein/β-actin for each sample.

### 2.10. Drugs

Monosodium urate, yohimbine, methysergide, and naloxone were purchased from Sigma Chemical Co. (St. Louis, MO, USA). Monosodium urate, yohimbine, methysergide, and naloxone were dissolved in saline. Rutin, luteolin-7-*O*-glucuronide, and apigenin-7-*O*-glucuronide were dissolved in saline. All drugs were prepared just before use.

### 2.11. Statistical Analysis

Statistical analysis was performed with the aid of GraphPad Prism Version 4.0 for Windows (GraphPad Software, San Diego, CA, USA). All values were expressed as the mean ± S.E.M. Data were compared between groups using one-way ANOVA, followed when necessary by the Bonferroni's post-hoc test. Differences were considered significant where $p < 0.05$.

## 3. Results

### 3.1. Identification of Isolated Compounds from Agrimonia pilosa Ledeb. Extract

Isolated compounds were analyzed by EI-MS (JEOL JMS-700, Freising, Germany), [1]H and [13]C-NMR (Bruker AV 400, Karlsruhe, Germany). The mass spectrometer was operated in the negative ion mode with the source, with electronic impact (EI) at 70 eV with a direct insertion probe. The ion source was set at 250 °C, and the mass range was 50–600 *m/z*. [1]H and [13]C-NMR spectra of the isolated pure compounds were recorded on a Bruker AV 400 instrument, using DMSO-$d_6$ as a solvent. In order to acquire UV spectra, we used the Finnigan Surveyor HPLC system (Thermo Electron, San Jose, CA, USA), which comprised of a PDA plus detector. The detailed structural information was listed as following (Figure 1)

**Compound 1:** [1]H-NMR (DMSO-$d_6$, 400 MHz) δ 7.56 (dd, J 9.0, 2.1 Hz, H-6'), 7.55 (d, J 2.1 Hz, H-2'), 6.85 (d J 8.4 Hz, H-5'), 6.40 (d, J 2.0 Hz, H-8), 6.11 (d, J 2.2 Hz, H-6), 5.35 (d, J 7.8 Hz, H-1''), 5.12 (d, J 1.6 Hz, H-1''') 3.83–3.35 (m, sugar-H), 1.13 (d, J 6.2 Hz, H-6''') [13]C-NMR (DMSO-$d_6$, 400 MHz) 178.2 (C-4), 164.3 (C-7), 161.5 (C-5), 157.1 (C-9), 156.7(C-2), 148.7 (C-4'), 145.6 (C-3'), 134.1 (C-3), 121.9 (C-1'), 122.0 (C-6'), 116.5 (C-5'), 116.1 (C-2'), 104.2 (C-10), 101.4 (C-glc-1), 101.0 (C-rha-1), 98.9 (C-6), 93.9 (C-8), 76.7 (glc-3), 76.7 (glc-5), 74.9 (glc-2), 72.7 (glc-4), 71.4 (rha-4), 71.2 (rha-3), 70.8 (rha-2), 68.5 (rha-5), 67.3 (glc-6), 18.0 (rha-6); EI-MS (*m/z*) 610 [M]$^+$, 465, 391,303; UV (MeCN, λ max nm) 361, 259. The [1]H-NMR, [13]C-NMR, MS, and UV data for compound 1 are identical to those reported previously [21]. Compound 1 was identified as rutin.

**Compound 2:** [1]H-NMR (DMSO-$d_6$, 400 MHz) $\delta$ 7.45 (dd, J 2.0, 8.0 Hz, H-6'), 7.44 (d, J 2.0 Hz, H-2'), 6.91 (d, J 8.3 Hz, H-5), 6.81 (d, J 2.0 Hz, H-8), 6.75 (s, H-3), 6.46 (d, J 2.2 Hz, H-6), 5.3 (d, J 7.3 Hz, H-1''), 4.1 (d, J 9.5 Hz, H-5''), 3.2–3.4 (m, H-2''-4''), [13]C-NMR (DMSO-$d_6$, 400 MHz) $\delta$ 180.9 (C-4), 173.1 (glu-6''), 165.4 (C-2), 163.2 (C-7), 160.9 (C-5), 153.1 (C-9), 153.9 (C-4'), 147.4 (C-3'), 119.8 (C-6'), 118.2 (C-1'), 116.5 (C-5'), 112.7 (C-2'), 104.9 (C-10), 101.9 (C-3), 101.1 (glu-1''), 99.8 (C-6), 93.9 (C-8), 76.88 (glu-3''), 73.9 (glu-5''), 73.3 (glu-2''), 72.1 (glu-4''); EI-MS (*m/z*) 461 [M − H]−, 285 [M − H-176]−; UV (MeCN, $\lambda$ max nm) 254, 266. The [1]H-NMR, [13]C-NMR, MS, and UV data for compound 2 are identical to those reported previously [22]. Compound 2 was identified as luteolin-7-*O*-glucuronide.

**Compound 3:** [1]H-NMR (DMSO-$d_6$, 400 MHz) $\delta$ 7.98 (d, J 8.7 Hz, H-2', 6'), 6.91 (d, J 8.7 Hz H-3', 5'), 6.83 (s, H-3), 6.81 (d, J 1.8 Hz, H-8), 6.42 (d, J 1.8 Hz, H-6), 5.12 (d, J 7.0 Hz, H-1''), 3.90 (d, J 9.7 Hz, H-5''), 3.2-3.5 (m, H-2''-4''), [13]C-NMR (DMSO-$d_6$, 400 MHz) $\delta$ 181.9 (C-4), 172.6 (glu-6''), 165.1 (C-2), 163.1 (C-7), 162.1 (C-5), 159.9 (C-4'), 157.6 (C-9), 129.3 (C-2',6'), 120.8 (C-1'), 116.7 (C-3', 5'), 105.9 (C-10), 102.8 (C-3), 99.5 (glu-1''), 100.2 (C-6), 95.3 (C-8), 77.2 (glu-3''), 74.4 (glu-5''), 73.7 (glu-2''), 72.3 (glu-4''); EI-MS (*m/z*) 447.18 [M]+, 269 [M-glu]+; UV (MeCN, $\lambda$ max nm) 267, 334. The [1]H-NMR, [13]C-NMR, MS, and UV data for compound 3 are identical to those reported in the literature [23]. Compound 3 was identified as apigenin-7-*O*-glucuronide. *glc: glucose; rha: rhamnose; glu: glucuronide.

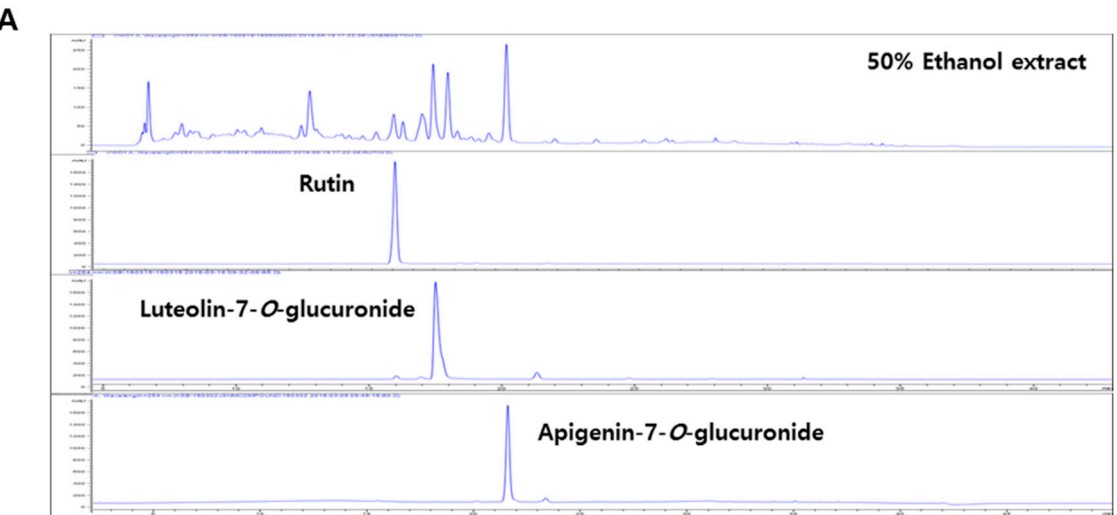

**Figure 1.** High Performance Liquid Chromatography (HPLC) patterns of the main composition of *Agrimonia pilosa* Ledeb. extract and chemical structure of isolated compounds. (**A**) The HPLC chromatogram analysis of isolated compound by Sephadex LH-20. Condition, column: Agilent Edipse XDB-Phenyl 4.6 × 150 mm (3.5 μm), mobile phase: solvent A (0.1% trifluoroacetic acid) and solvent B (acetonitrile) in gradient mode, flow-rate: 0.7 mL/min, detection at 254 nm. (**B**) Chemical structure of isolated components from *Agrimonia pilosa* Ledeb.

### 3.2. Effect of Single Components Isolated from Agrimonia pilosa Ledeb. Extract on Mechanical Pain Threshold in MSU-Treated Pain Model

As shown in Figure 2, three single components from AP extract were selected, and the possible antinociceptive effects of those components were assessed. These three single components (rutin, luteolin-7-*O*-glucuronide, and apigenin-7-*O*-glucuronide) were administered orally and caused dose-dependent reversals of reduced pain threshold in the MSU-induced pain model, as shown in Figure 2. Among the 3 components, apigenin-7-*O*-glucuronide was more effective in the production of antinociception.

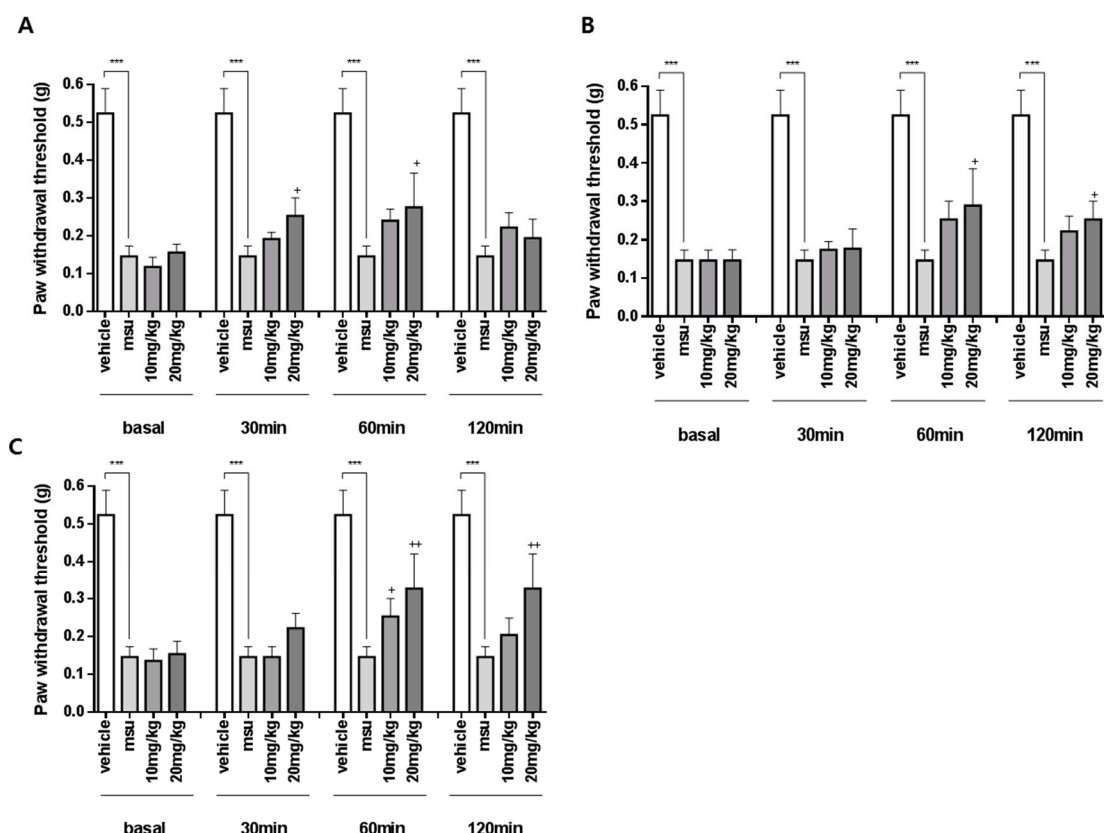

**Figure 2.** The antinociceptive effect of rutin, luteolin-7-*O*-glucuronide, and apigenin-7-*O*-glucuronide administered orally in MSU-induced pain model. Mice were administered orally with (**A**) rutin, (**B**) luteolin-7-*O*-glucuronide, or (**C**) apigenin-7-*O*-glucuronide (10 or 20 mg/kg) and the pain thresholds were measured at 30, 60, and 120 min after treatment using Von-Frey test. The vertical bars denote the standard error of the mean. $n = 5$ per group. (**A**) *** $p < 0.001$ (vehicle vs. MSU), + $p < 0.05$ (MSU vs. 20 mg/kg). (**B**) *** $p < 0.001$ (vehicle vs. MSU), + $p < 0.05$ (MSU vs. 20 mg/kg). (**C**) *** $p < 0.001$ (vehicle vs. MSU), + $p < 0.05$ (MSU vs. 10 mg/kg), ++ $p < 0.01$ (MSU vs. 20 mg/kg).

### 3.3. Effect of Apigenin-7-O-Glucuronide on Pain Behavior in Writhing and Formalin Tests

As shown in Figure 3A, the mean number of abdominal constrictions in vehicle-treated control animals was approximately 36. Oral administration with apigenin-7-*O*-glucuronide suppressed the acetic acid-induced writhing numbers in a dose-dependent manner. In the vehicle-treated control group, 5% formalin was injected to cause acute pain response (first phase) that lasted for 5 minutes, and chronic inflammatory response (second phase) which began about 20 min after formalin administration and lasted for about 20 min (Figure 3B). In apigenin-7-*O*-glucuronide-treated groups, the doses of 10 and 20 mg/kg significantly reduced the paw-licking/biting response time only in the second phase of formalin nociception, but not the first phase (Figure 3B).

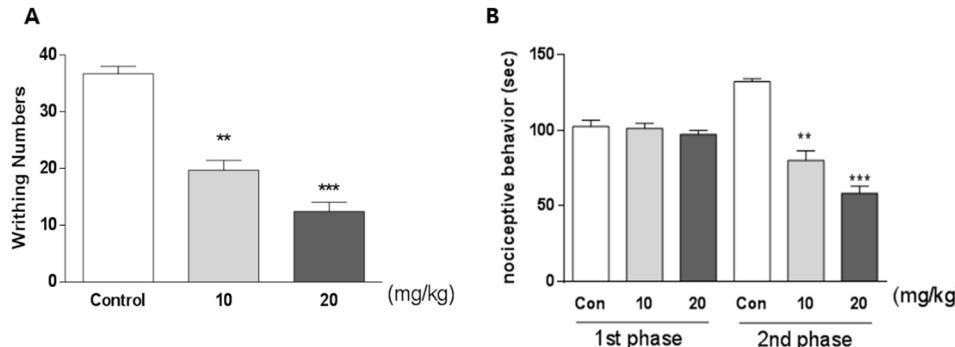

**Figure 3.** Effect of apigenin-7-*O*-glucuronide on the nociceptive response induced by acetic acid and formalin. (**A**) Apigenin-7-*O*-glucuronide (10 or 20 mg/kg) was administered orally 30 min prior to the acetic acid (1%, 250 μL) injection. The number of writhing was counted for 30 min following acetic acid injection. (**B**) Mice were administered orally with apigenin-7-*O*-glucuronide (10 or 20 mg/kg) for 30 min prior to the formalin (5%, 10 μL) injection. The cumulative response time was measured during the first phase (0–5 min) and second phase (20–40 min). The vertical bars indicate the standard error of the mean. $n = 8$–$10$ per group (** $p < 0.01$, *** $p < 0.001$, compared with control group).

### 3.4. Effect of Serotonergic, Adrenergic, and Opioidergic System on the Inhibition of Writhing Response Induced by Apigenin-7-O-Glucuronide Administered Orally

To further determine the possible role of spinal serotoninergic, adrenergic, and opioid systems in apigenin-7-*O*-glucuronide-induced antinociception, we pretreated these receptor blockers with i.t., respectively, and 1% acetic acid was administered by i.p. injection. Pretreatment with yohimbine (an α2-adrenergic receptor blocker) (Figure 4C), but not methysergide (a serotonergic receptor antagonist) (Figure 4A), or naloxone (an opioidergic receptor antagonist) (Figure 4B), attenuated apigenin-7-*O*-glucuronide-induced inhibition of writhing response. Furthermore, the treatment of methysergide, yohimbine or naloxone itself had no influence on the writhing response (Figure 4).

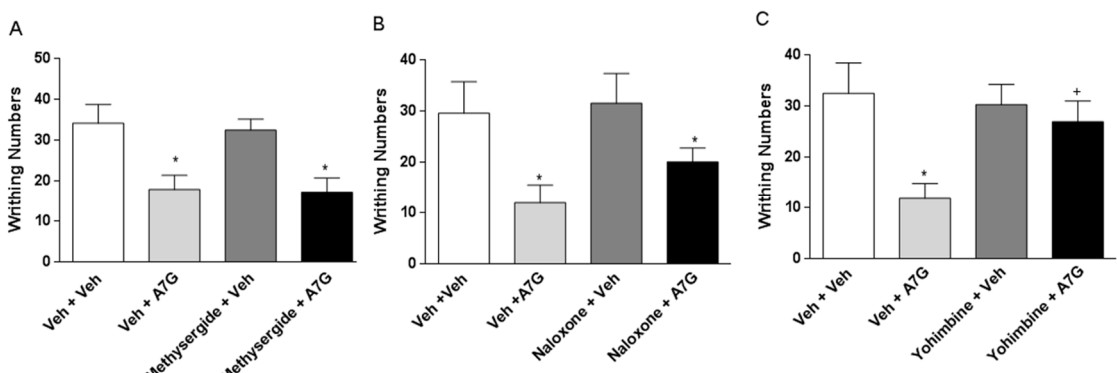

**Figure 4.** Effect of methysergide (**A**), yohimbine, (**B**) and naloxone (**C**) pretreated intrathecally (i.t.) on inhibition of the antinociception induced by apigenin-7-*O*-glucuronidein the writhing test. Methysergide, yohimbine, or naloxone (0.01 μg/5 μL) was pretreated i.t. for 5 min. Mice were administered orally with apigenin-7-*O*-glucuronide (20 mg/kg) for 30 min prior to the i.p. injection with 1% acetic acid. The number of writhing was counted for 30 min following acetic acid injection. The vertical bars indicate the standard error of the mean. $n = 8$–$10$ per group (* $p < 0.05$, compared with control group).

### 3.5. Changes of Phosphorylated mTOR, P38, and CREB Proteins in the Spinal Cord by Apigenin-7-O-Glucuronide in the Formalin Test

Spinal expression of mTOR, P38, or CREB protein phosphorylation was evaluated by western blotting. The lumbar spinal cord was dissected at 20 min after formalin injection. Formalin injection

caused upregulation of p-mTOR, p-P38, and p-CREB expression in the spinal cord. In addition, oral administration with apigenin-7-O-glucuronide (20 mg/kg) attenuated formalin-induced p-CREB, p-P38, and p-mTOR levels (Figure 5).

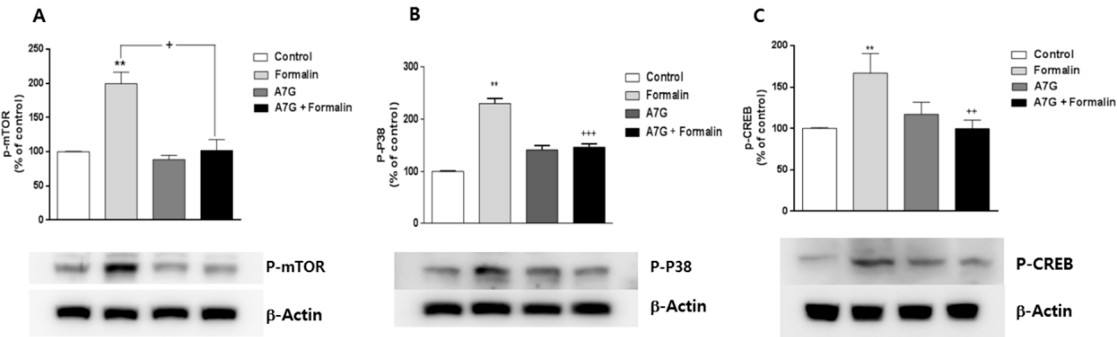

**Figure 5.** Changes of phosphorylated ERK, P38, and CREB proteins in the Spinal cord by apigenin-7-O-glucuronidein in the formalin test. β-Actin was used as an internal control. Values are mean ± SEM. These values were expressed as the percentage of the control tested protein/β-actin for each sample. $n = 6$ per group. (* $p < 0.05$, ** $p < 0.01$, compared with Control group; + $p < 0.05$, ++ $p < 0.01$, +++ $p < 0.001$, compared with Formalin group).

## 4. Discussion

The finding of this study revealed that three single components, rutin, luteolin-7-O-glucuronide, and apigenin-7-O-glucuronide, isolated from AP orally administered at the doses of 10 and 20 mg/kg markedly produce an antinociceptive effect in monosodium urate (MSU)-induced pain model, of which apigenin-7-O-glucuronide was more effective in the production of antinociception. Additionally, oral administration apigenin-7-O-glucuronide suppressed the writhing number induced by 1% acetic acid and inhibited the formalin-induced nociception during the second phase, which confirmed the antinociceptive effect of this formulation.

Several previous studies have demonstrated that MSU is used as gout pain model [24,25]. In the present study, we confirmed that MSU treatment caused a reduction of mechanical pain threshold as measured by a Von Frey stimulation of plantar of the mouse. Oral administration of rutin, luteolin-7-O-glucuronide, and apigenin-7-O-glucuronide causes the reversal of decreased threshold of MSU-induced pain, suggesting that these three components isolated from AP extract exert antinociceptive effects in MSU-induced pain. In addition, among 3 components, apigenin-7-O-glucuronide was more effective than rutin and luteolin-7-O-glucuronide in the MSU-induced pain model. Our results were in part in line with previous findings in that rutin and luteolin-7-O-glucuronide show antinociceptive effect in pain models. They exhibit a wide variety of biological activities, including antiviral, antibacterial, anti-inflammatory, anti-carcinogenic, and antioxidant actions [26–28]; and also exerts an antinociceptive effect in neuropathic pain model and anti-arthritic activity [29–31]. However, no antinociceptive effect by apigenin-7-O-glucuronide has been previously reported. Thus, we further tried to characterize of apigenin-7-O-glucuronide-induced antinociceptive effect. We found that apigenin-7-O-glucuronide was also effective to reduce pain behavior as revealed in the writhing and formalin tests, indicating that apigenin-7-O-glucuronide reduces chemical-induced nociception.

Interestingly, we examined the analgesic effect of aspirin and acetaminophen in a previous study, when the potency was compared, apigenin-7-O-glucuronide showed a better potency compared to that of aspirin and acetaminophen. In our previous study, aspirin or acetaminophen at doses of 200 and 300 mg/kg significantly attenuated pain behavior induced by TNF-α, IL-1β, or IFN-γ administered intrathecally [32]. However, in the present study, apigenin-7-O-glucuronide reduced antinociception in acetic acid and formalin-induced inflammatory models only at doses of 10 and 20 mg/kg. Based upon these results, apigenin-7-O-glucuronide isolated from AP may be applied as an effective analgesic drug.

Several pain inhibitory systems such as opioid, serotonin, and norepinephrine systems spinally located are known to play important roles in the regulation of pain or antinociception. Opioid receptors are involved in controlling many regions of the nervous system [33]. Our results shows that naloxone, a classic opioid receptor antagonist, could not antagonize the antinociception of apigenin-7-*O*-glucuronide in the writhing test. It indicates that the opioid receptor is not involved in the antinociception of apigenin-7-*O*-glucuronide. However, several studies reported that the opioid receptor could be regulated by descending serotonergic and noradrenergic pain inhibitory systems [33,34]. In the present study, we observed that the yohimbine (a blockade of spinal α2-adrenergic receptors) significantly reduce apigenin-7-*O*-glucuronide-induced antinociception; whereas, methysergide (the blockade of serotonergic receptors) appears no effect on the antinociception of apigenin-7-*O*-glucuronide. The present study suggested that orally administered apigenin-7-*O*-glucuronide-induced antinociception may be mediated by the activation of spinal α2-adrenergic receptors instead of spinal opioidergic and serotonergic receptors. The result was also supported by a previous study, which confirmed that the α2-adrenergic receptors, but not opioidergic or serotonergic receptors, are involved in AP extract-induced antinociception [16].

Recently, several series of studies reported that p-mTOR and p-P38 expression were altered in the spinal cord or dorsal root ganglia in the neuropathic pain model [35,36]. In addition, in the spinal cord or brain regions, p-CREB and p-mTOR proteins were increased in an acute inflammatory pain model, such as the formalin pain model [37,38]. In support of these findings, in the present study, we found that p-mTOR, p-P38, and p-CREB expressions are up-regulated in formalin-induced pain model. Besides, we found that pretreatment with apigenin-7-O-glucuronide almost abolished the p-mTOR, p-P38, and p-CREB expressions induced by formalin. Our findings indicate that the suppression of nociception by apigenin-7-*O*-glucuronide treatment in the formalin pain model might be mediated by the reductions of p-mTOR, p-P38, and p-CREB level in the spinal cord. This finding at least could explain the antinociceptive action of this flavonoid in like inflammatory pain.

Although the exact peripheral antinociceptive mechanism of apigenin-7-*O*-glucuronide was not examined in the present study, it is speculated that apigenin-7-*O*-glucuronide on the peripherally may have anti-inflammatory effect. Hu et al. have shown that apigenin-7-*O*-glucuronide exerted an anti-inflammatory effect in LPS-induced inflammation in RAW 264.7 cells via inhibiting LPS-induced proinflammatory cytokines [39]. Several inflammatory mediators, such as nitric oxide (NO), tumor necrosis factor (TNF)-α, interleukin (IL)-1β, and interleukin 6 (IL-6), have been reported to be involved in the inflammatory response [39]. And this reaction requires upregulated activity of mitogen-activated protein kinases (MAPKs), including extracellular signal-regulated kinase (ERK), c-Jun N-terminal kinase (JNK), and p38. Activation of these signaling molecules is linked to the activation of transcription factors such as nuclear factor (NF)-κB, cAMP response element-binding protein (CREB) [39]. Consistent with these results, we inferred that apigenin-7-O-glucuronide effectively reduced nociceptive responses via downregulation of inflammatory-related gene expression through the suppression of MAPK signaling pathways in formalin-induced inflammation.

## 5. Conclusions

In conclusion, the results of this study showed that three single components (rutin, luteolin-7-*O*-glucuronide, and apigenin-7-*O*-glucuronide) isolated from AP pain model have antinociceptive effects, as confirmed by behavioral tests in MSU-induced pain model. Apigenin-7-*O*-glucuronide had a significant effect on the reduction of nociceptive response in acetic acid-induced pain model via spinal α2-adrenergic receptors, as well as in a formalin-induced pain model by the reductions of the expression of p-P38, p-CREB, and p-mTOR levels in the spinal cord.

**Author Contributions:** Conceptualization, S.S.L. and H.W.S.; Formal analysis, J.H.F., H.J.L. and S.B.K.; Investigation, J.H.F., H.J.L. and J.S.J.; Project administration, S.S.L. and H.W.S.; Resources, H.J.L. and S.B.K.; Visualization, J.H.F., H.J.L. and S.B.K.; Writing–original draft, H.W.S.; Writing–review & editing, J.H.F.

**Funding:** This work was supported by Hallym University Fund (HRF-201807-011).

**Conflicts of Interest:** The authors declare no conflict of interest. The funders had no role in the design of the study; in the collection, analyses, or interpretation of data; in the writing of the manuscript, or in the decision to publish the results.

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
