# Peer review of "Antinociceptive Effect of Single Components Isolated from Agrimonia pilosa Ledeb. Extract"

_scipharm, doi:10.3390/scipharm87030018_

Reviewer 1 Report

The article entitled “Antinociceptive effect of single components isolated from Agrimonia pilosa Ledeb extract” by Jing Hui Feng, et al., has found some interesting results, however, some questions should be solved before making a decision.

Major questions

1.      It is confused about the abstract. What is the relationship between Agrimonia pilosa Ledeb (AP) extract and apigenin-7-O-glucuronide? Authors should rewrite the abstract.

2.      Authors should rewrite the paragraph of introduction. Authors should describe more details about “these” compounds mentioned in the abstract. Dose apigenin-7-O-glucuronide from Agrimonia pilosa Ledeb (AP) extract? Why choose yohimbine or naloxone or methysergide? What is the purpose of this article?

3.      Authors should describe more details about Von-Frey test.

4.      Why choose the doses of 10 and 20 mg/kg of apigenin-7-O-glucuronide?

5.      Authors should explain why test the proteins of mTOR, p-38, and p-CREB? What is the roles of these proteins in the pain model in this study?

6.      The manuscript should be proofread thoroughly for errors in grammar and spellings.

Author Response

Response to Reviewer 1 Comments

Point 1.  It is confused about the abstract. What is the relationship between Agrimonia pilosa Ledeb (AP) extract and apigenin-7-O-glucuronide? Authors should rewrite the abstract.

Response 1: Thank you for your kind consideration. We have rewritten the abstract (Please see Page 1, Line 14-21).

Point 2.  Authors should rewrite the paragraph of introduction. Authors should describe more details about “these” compounds mentioned in the abstract. Dose apigenin-7-O-glucuronide from Agrimonia pilosa Ledeb (AP) extract? Why choose yohimbine or naloxone or methysergide? What is the purpose of this article?

Response 2: Thank you for your kind suggestion. Apigenin-7-O-glucuronide from is isolated from AP. We have rewritten the abstract (Please see Page 1, Line 14-21).

    The present study was designed to examine the possible effects of several single components isolated from AP extract in various pain models. In addition, we have tried to find the pharmacological and molecular mechanisms of those components in the production of antinociception. So, we choose yohimbine (an α2-adrenergic receptor blocker), methysergide (a serotonergic receptor antagonist) or naloxone (an opioidergic receptor antagonist) to find the pharmacological mechanisms of A7G-induced antinociceptive effect (Please see Page 9, Line 297-311).

Point 3.  Authors should describe more details about Von-Frey test.

Response 3: Thank you for your kind suggestion. We have added some details about Von-Frey test in the manuscript (Please see Page 3, Line 102-108).

Point 4.  Why choose the doses of 10 and 20 mg/kg of apigenin-7-O-glucuronide?

Response 4: Thank you for your kind suggestion. 10 and 20 mg/kg of apigenin-7-O-glucuronide were selected by speculation that the analgesic effect of apigenin-7-O-glucuronide should be pharmacologically proved in a dose-dependent manner.

Point 5.  Authors should explain why test the proteins of mTOR, p-38, and p-CREB? What is the roles of these proteins in the pain model in this study?

Response 5: Thank you for your kind suggestion. mTOR, p-38, and p-CREB proteins were already well known that they exert important functional roles during the pain transmission in formalin-induced pain model as described in the Discussion Section. Therefore, we wanted to examine if the compound was effective to affect formalin-induced phosphorylation of mTOR, p-38, and p-CREB proteins. Also, we added some information about mTOR, p-38, and p-CREB proteins in the Introduction Section. (Please see Page 2, Line 52- 61; Page 9, Line 312-334).

Point 6.  The manuscript should be proofread thoroughly for errors in grammar and spellings.

Response 6: Thank you for your kind suggestion. (Page 3, Line 132; Page 7, Line 246).

Please see the attachment. We uploaded the revised manuscript file.

Reviewer 2 Report

The authors have presented an interesting study on the antinociceptive effect of isolated compounds from Agrimonia pilosa extract. The results of the study could improve the understanding of the complex mechanisms of action of natural compounds. However, a few issues in the manuscript need to be further discussed or corrected:

The authors should add to the discussion more information concerning a possible peripheral antinociceptive effect of A7G. The writhing model is well known for the involvement of peripheral mediators of inflammation and pain, thus a possible influence of A7G on these mediators should not be excluded even though it was not investigated by the authors.

The authors should explain why they have not used a reference antinociceptive drug in their studies, it would have been interesting to compare the magnitude of the effect of A7G with an existing analgesic drug.

In the MSU-induced gout model, the authors have not explained which limb was tested by Von Frey method: only the injected one or also the contralateral limb.

The authors should check again the reference list, for example at line 262, reference [19] cannot explain MSU model since it is about a manuscript from 9th century. It was Sir Alfred Baring Garrod who discovered the key role of uric acid in gout but only in the 19th century.

A few minor errors need correction: line 96 intraplantar instead of intraplanar; line 139 yohimbine instead of yohombine.

In Introduction, line 40 the abbreviation AP should be given its full name, it is for the first time mentioned in the text, regardless of the abstract.

Author Response

Response to Reviewer 2 Comments

Point 1. The authors should add to the discussion more information concerning a possible peripheral antinociceptive effect of A7G. The writhing model is well known for the involvement of peripheral mediators of inflammation and pain, thus a possible influence of A7G on these mediators should not be excluded even though it was not investigated by the authors.

Response 1: Thank you for your kind suggestion. We added more information concerning a possible peripheral antinociceptive effect of A7G in the Discussion Section. (Please see Page 9, Line 322-334).

Point 2. The authors should explain why they have not used a reference antinociceptive drug in their studies, it would have been interesting to compare the magnitude of the effect of A7G with an existing analgesic drug.

Response 2: Thank you for your kind suggestion. Although we have not examined a reference antinociceptive drug in the present study, we have examined the analgesic effect of aspirin and acetaminophen in a previous study. We added some information in the Discussion Section. (Please see Page 9, Line 314-326).

Point 3. In the MSU-induced gout model, the authors have not explained which limb was tested by Von Frey method: only the injected one or also the contralateral limb.

Response 3: Thank you for your kind suggestion. In the Von Frey test, we measured the pain threshold of the right hind paw of the mouse. And we have made changes in the manuscript (Please see page 3, line 102).

Point 4. The authors should check again the reference list, for example at line 262, reference [19] cannot explain MSU model since it is about a manuscript from 9th century. It was Sir Alfred Baring Garrod who discovered the key role of uric acid in gout but only in the 19th century.

Response 4: Thank you for your kind suggestion. We checked the reference list, and we changed reference [26] (the list of references uploaded) (Please see page 11, line 417-419).

Point 5. A few minor errors need correction: line 96 intraplantar instead of intraplanar; line 139 yohimbine instead of yohombine.

Response 5: I am sorry that we made these mistakes. We corrected these errors by converting “intraplanar” into “intraplantar” (Please see page 3, line 108) and converting “yohombine” into “yohimbine” (Please see page 4, line 151 and 152).

Point 6. In Introduction, line 40 the abbreviation AP should be given its full name, it is for the first time mentioned in the text, regardless of the abstract.

Response 6: As you have suggested, we added the full name of the abbreviation AP in the Introduction Section (Please see page1, line 38).

Please see the attachment. We uploaded the revised manuscript file.

Round  2

Reviewer 1 Report

The article entitled “Antinociceptive effect of single components isolated from Agrimonia pilosa Ledeb extract” by Jing Hui Feng, et al., did revise the comments from reviewer.

Reviewer 2 Report

I am glad that my suggestions have been taken into consideration by the authors, therefore I recommend the publication of the revised manuscript.

This manuscript is a resubmission of an earlier submission. The following is a list of the peer review reports and author responses from that submission.